# DEMIX: Domain-Enforced Memory Isolation for Embedded System [note 1]

**DOI:** 10.3390/s23073568

**Published:** 2023-03-29

**Authors:** Haeyoung Kim, Harashta Tatimma Larasati, Jonguk Park, Howon Kim, Donghyun Kwon

**Affiliations:** 1Information Security and AIoT Laboratory, School of Computer Science & Engineering, Pusan National University, Busan 46241, Republic of Koreahowonkim@pusan.ac.kr (H.K.); 2Computer Security Laboratory, School of Computer Science & Engineering, Pusan National University, Busan 46241, Republic of Korea

**Keywords:** memory isolation, embedded systems, risc-v

## Abstract

Memory isolation is an essential technology for safeguarding the resources of lightweight embedded systems. This technique isolates system resources by constraining the scope of the processor’s accessible memory into distinct units known as domains. Despite the security offered by this approach, the Memory Protection Unit (MPU), the most common memory isolation method provided in most lightweight systems, incurs overheads during domain switching due to the privilege level intervention. However, as IoT environments become increasingly interconnected and more resources become required for protection, the significant overhead associated with domain switching under this constraint is expected to be crucial, making it harder to operate with more granular domains. To mitigate these issues, we propose DEMIX, which supports efficient memory isolation for multiple domains. DEMIX comprises two mainelements—*Domain-Enforced Memory Isolation* and *instruction-level domain isolation*—with the primary idea of enabling granular access control for memory by validating the domain state of the processor and the executed instructions. By achieving fine-grained validation of memory regions, our technique safely extends the supported domain capabilities of existing technologies while eliminating the overhead associated with switching between domains. Our implementation of eight user domains shows that our approach yields a hardware overhead of a slight 8% in Ibex Core, a very lightweight RISC-V processor.

## 1. Introduction

In the era of the Internet of Things (IoT), the utilization of embedded systems has become highly prevalent, ranging from its use in personal health care [1], home automation [2], intelligent transportation system and autonomous vehicles, to the large-scale industry [3,4]. Nevertheless, their lightweight characteristics have made it difficult to equip them with strong security mechanisms applicable to their high-end systems counterparts. In particular, these embedded systems are often programmed in unsafe language such as C/C++, rendering them susceptible to grave security threats such as code injection, buffer overflow, and memory leaks. In fact, it has been shown that various embedded systems have been compromised through these security vulnerabilities [5,6,7,8], which may lead to potentially fatal consequences. Due to a lack of architectural support, established security solutions for advanced computing systems such as PCs and smartphones are not directly transferable to embedded systems. For instance, hardware virtualization extensions [9,10], SGX enclaves [11,12,13], and tagged memory architectures [14,15,16], which are commonly used in security solutions, are unsuitable for low-end embedded devices due to the additional hardware costs and power consumption. To overcome this challenge, researchers have proposed alternative methods to enhance the security of embedded systems, such as the memory isolation technique [17,18,19,20]. This technology enhances the security of embedded systems by limiting the range of accessible code and data regions. The memory isolation approach effectively secures the system resources of individual processes by protecting access to unauthorized memory regions from software vulnerabilities.

The most common method of providing memory isolation by domain in embedded systems is using a Memory Protection Unit (MPU). The MPU enables the definition of memory regions and their associated permissions, thus constraining the scope of the processor’s accessible memory into distinct domains. However, despite the memory isolation provided by the MPU, a trade-off exists in that it incurs overheads during domain switching due to the privilege level intervention. This overhead can lead to latency issues if domain switching occurs frequently. To address this limitation, several studies have proposed ways to support unprivileged domains, such as EA-MPU [18,19], ARM Trustzone-M [21]. These approaches eliminate the intervention of the privilege level during the domain-switching process, thus reducing the overhead associated with this conversion process. However, as the number of unprivileged domains increases, so does the complexity of the domain protection function. Most proposals have handled this complexity by using specialized domains for switching or simplifying domains by providing only secure domains to protect sensitive resources.

Our previous work, RIMI [22], is an approach that proposed to support unprivileged domains by dividing each memory region into two domains and implementing instruction-level domain isolation to provide dedicated instructions for accessing each part. This method offers fine-grained access control by distributing access permissions to code in an instruction unit. However, RIMI focuses primarily on protecting the two domains for data, with limited consideration given to protecting the domain’s code. Furthermore, similar to other technologies, RIMI only supports two domains, and authorized validation is based only on instructions. As a result, its scalability is limited, as defining multiple domains required adding instruction sets equal to the number of domains to be supported.

In this paper, we present DEMIX, our extended approach that addresses the limitations of existing methods for supporting unprivileged domains. DEMIX expands the number of supported domains while eliminating the overhead associated with domain switching, enabling more efficient use of resources with providing safety in lightweight embedded systems. In particular, DEMIX consists of two components: (1) *Domain-Enforced Memory Isolation*, which enables supporting multiple domains and eliminating the overhead required to switch between each domain, and (2) *instruction-level domain isolation*, which addresses permission issues that arise as the number of supported domains increases, and prevents domain fragmentation for internal data protection.

In detail, our Domain-Enforced Memory Isolation technique serves as an accessible memory region through a separate domain state register managed by the processor. This register, acting as a domain identifier, allows quick reconfiguration of the address space upon domain switching by changing the stored value. Meanwhile, our instruction extension extends the instruction-level memory isolation proposed previously in RIMI. However, unlike the previous proposal that only supports two domains, our domain protection is now developed for multiple domains, which is achieved by combining it with our Domain-Enforced Memory Isolation technique. Furthermore, we also incorporate a new instruction to handle domain switching permissions issues in multiple domains support. This instruction designates domain-specific accessible entrances to code regions when a processor switches domains. As a result, our technology supports instantaneous switching for multi-domain while offering enhanced protection during domain switching. Implemented on a RISC-V core that does not support unprivileged domains, our evaluation demonstrates that DEMIX is able to support unprivileged domains with minimal resource overhead on lightweight embedded systems. More importantly, our evaluation provides evidence of the feasibility and effectiveness of our approach in improving the security of lightweight embedded systems.

The advantages of DEMIX over existing solutions can be summarized as follows:**Increased number of supported domains:** DEMIX supports multiple domains, unlike some proposals that only support one secure environment. This feature makes it more scalable and better suited for real-world use cases that are more likely to require multiple domains.**Reduced overheads:** DEMIX eliminates the overhead associated with domain switching, allowing for more efficient use of resources. It includes the availability of processors and the consumption of MPU entries required to use a separate switching domain.**Enhanced protection during domain switching:** DEMIX ensures granular permissions and correct access points per unit task inside the domain, providing improved security while minimizing the risks associated with cross-domain function calls.**Fine-grained access control:** DEMIX offers fine-grained access control by distributing access permissions to code in an instruction unit, providing an added layer of security for embedded systems. This capability also defines highly distributed or heterogeneous privilege configurations with minimal MPU entries.**Compatibility with lightweight processors:** DEMIX has been successfully implemented on the Ibex processor, a lightweight RISC-V core. This makes DEMIX a practical solution in real-world IoT environments with prevalent lightweight embedded systems.

This paper is organized as follows. First, we provide the motivation in Section 1 as well as the background and related works for the proposed memory isolation technique in Section 2. Subsequently, we outline our assumptions, threat models, and the design requirements in Section 3. Next, we present a detailed description of our proposed DEMIX in Section 4, before presenting the implementation of DEMIX on Reduced Instruction Set Computer version 5 (RISC-V) architecture [23] and analyzing the results in Section 5. After assessment and performance comparison with related work, we discuss limitations and future work in Section 6.

## 2. Background and Related Work

Prior to our research, there have been several works that aim to implement unprivileged domains in embedded systems. These studies have utilized various methods to separate address spaces and enforce memory protection. In this section, we provide a brief review of these existing approaches.

### 2.1. Memory Protection Unit

The Memory Protection Unit (MPU) [24,25], one of the most popular techniques to provide memory isolation by domain in embedded systems, is commonly implemented as a dedicated hardware component and manages memory access permissions and restrictions. However, this method comes with the drawback of overheads when switching domains. Specifically, the cost of switching privilege levels incurs as MPUs can only be configured in the privileged mode for domain protection. Additionally, the cost of reconfiguring the MPU is also incurred during domain switching. These overheads can be significant for applications that frequently change domains.

To improve the efficiency of MPU, some works have proposed an extended MPU called the Execution-Aware MPU (EA-MPU) [18,19]. EA-MPU proceeds further than traditional MPUs by managing the legitimate code regions that can be accessed for each memory entity. This functionality enables protection against unauthorized memory access from malicious code execution. Furthermore, the EA-MPU approach can reduce the overhead of address space reconfiguration by allowing access permissions to change based on execution location without privilege-level intervention. However, the limited number of memory protection regions supported by most lightweight embedded devices makes it difficult to set up distributed memory regions. It is challenging to control fine-grained authority based on the data’s role or the unit task’s functionality for higher security. In particular, because the technology uses entity tables to provide cross-domain function calls, configuring domain-specific permissions requires as many entity tables as supported domains.

### 2.2. Memory Protection Key

The Memory Protection Key (MPK) is a hardware-based memory protection technology that provides a set of hardware-enforced protection keys. MPK checks the protection key associated with a memory region and grants access if the key matches the desired value. This method provides a separate identifier for validating the domain, allowing for a fast transition by changing the protection key or a designated register instead of reconfiguring the entire address space. Intel MPK [26] is an example of this technology, which also allows for the division of virtual pages into up to 16 domains. Next, read and write permissions for each page are controlled by the Protection Key Rights Register (PKRU), in which the user program can modify. This capability provides a low overhead for changing address space. However, Intel MPK is designed for high-end systems and may be vulnerable to attacks that exploit weaknesses because it allows an adversary to control read and write permissions for each domain [27,28,29].

Another example of using MPK technology is the Domain Keys Efficient In-Process Isolation for RISC-V and x86 (Donky) [30], which also applies this technology to a RISC-V processor. Donky addresses the limitation of the existing RISC-V PMP (i.e., no support for user-level domain isolation) by incorporating the MPK extension. Unlike other techniques, Donky provides a so-called Donky Monitor that runs in user space without the need for kernel or privilege-level intervention, and a hardware call gate (dcall) to invoke it. This reduces domain switching overhead and provides fine-grained access control in environments that require frequent domain switching. However, the need for monitor calls during the domain-switching process incurs overhead, even though the monitor operates at an unprivileged level. As a result, Donky is useful for traditional context-based protection but may not be ideal for providing more granular control over resources within a single context.

### 2.3. Horizontal Partitioning

Horizontal partitioning is a security technique that divides data and code into separate partitions. This method offers a minimal number of domains and policies, usually just two, but is optimized for applications that require fast domain switching. TrustZone-M [21] is a security extension of the ARM architecture designed specifically for embedded systems. It separates system resources, including memory and peripherals, into two separate domains known as secure and normal words. Despite the limitations in flexibility and scalability posed by the fixed domains provided by TrustZone-M, it allows switching between domains without needing privilege-level intervention. To restrict access to the secure world from the normal world, a Non-Secure Callable (NSC) region is provided, which comprises a table of entries. This approach faces similar challenges to traditional MPU-based approaches, where it is difficult to accurately grant access to a specific data region to highly distributed locations of code.

### 2.4. RIMI—Preliminary Version of DEMIX

Our initial version of this work, RIMI [22], partitions code and data into two domains and provides separate instructions for accessing each domain, ensuring that the designated instructions can only access their designated areas. This enables the creation of fine-grained, highly distributed permissions within the instruction unit. Even though RIMI provides independent instruction-level protection for code and data regions, it supports only two limited domains. Consequently, RIMI can grant access to highly dispersed areas, such as Shadow Stack [31,32], or control memory access according to the code domain, but not both simultaneously. Furthermore, even though access to the opposite domain is only achievable through dedicated instructions, it cannot restrict access to specific entry points in the code. This means that all code space of the opposite domain can be accessed using the dedicated instructions.

## 3. Design Considerations

In this section, we elaborate on the design considerations of DEMIX, our proposed novel Domain-Enforced Memory Isolation technique that offers an efficient and secure solution for safeguarding sensitive code and data in embedded systems. Our proposal aims to address the challenges that lightweight embedded processors encounter in protecting their memory against malicious attacks by combining the benefits of existing methods while also overcoming their limitations. The primary goal of DEMIX is to provide strong memory isolation in embedded systems, thereby enhancing the overall security of these systems. In detail, we first describe the threat model, assumptions, and challenges to implementing DEMIX. Subsequently, we elaborate on how DEMIX handles these aspects, providing a clear mapping of the features and the corresponding threat and challenges it addresses.

### 3.1. Threat Model and Assumptions

There are a variety of attacks that can be executed by exploiting the vulnerability of a system, in which the scenarios can be described with different threat models as presented in [33]. In terms of embedded systems, one of the solutions to safeguard against these attack memory isolation using the MPU, which can provide device-oriented privacy by limiting the area of code and data accessible to the processor. Each software component is isolated and can only access other resources of the embedded system with extra permission [33]. These separated areas can be referred to as domains, and the process of changing these domains is mostly conducted through privilege intervention.

In this paper, we propose an efficient unprivileged domain isolation technique to safeguard internal resources in lightweight embedded systems. Each domain that we design protects internal resources without requiring privilege-level intervention and supports instant transitions. Although our proposal primarily focuses on unprivileged-level isolation, these domains are still protected through the privileged level. Therefore, we follow the same assumptions as the existing memory isolation approach. In particular, we assume a trusted software component that operates as a Trusted Computing Base (TCB) in the system exists. This TCB ensures the integrity of critical resources for the unprivileged domain of DEMIX, including the Memory Protection Unit (MPU) configuration. We also assume that memory vulnerabilities exist in the embedded system’s software and a remote attacker has knowledge of them and can exploit them. The adversary aims to break the security by obtaining unauthorized access to sensitive functions or leaking critical data in the embedded system.

Memory isolation that does not support multiple unprivileged domains can prevent this unauthorized access using privilege-level software. This software is always performed when switching domains, configuring the new memory access scope on the MPU, and validating legitimate access. However, as we support unprivileged domains that remove the overhead of privilege-level intervention in these processes, some threats need to be considered. Figure 1 illustrates the threats in the unprivileged domain, which arise from removing privilege boundaries, denoted as T1, T2, T3, and T4, described below.

**T1.** **Unauthorized Access to Sensitive Memory:** Unauthorized access to sensitive memory (e.g., code and data) is a significant concern in the systems that implement unprivileged domains and utilize MPU to manage memory access permissions for multiple domains. An adversary may attempt to gain access to sensitive data and functions from other domains that are accessible at an unprivileged level, potentially compromising the security of the system.**T2.** **Malicious Cross-Domain Function Calls:** Malicious cross-domain function calls can pose a significant security risk through incomplete function execution. An adversary may attempt to exploit the cross-domain function call mechanism by branching to a location that is not a valid function entrance for a trusted domain. This may result in unauthorized access to sensitive data or functionality, compromising the system’s security.**T3.** **Tampering with Domain Switching Process:** An adversary may attempt to gain unauthorized access to sensitive data and functionality by compromising memory isolation by manipulating the domain-switching process. This can be achieved by posing as a different domain, potentially bypassing security measures and compromising the system’s security.**T4.** **Exploiting Shared Memory:** Shared memory can be a necessary component for cross-domain data communication, but it can also be a target of exploitation by malicious parties. An adversary may attempt to compromise sensitive information or secure contexts of other domains by modifying their addresses to a shared memory location, potentially leading to data leaks.

Note that we focus on providing a reliable and flexible solution for memory protection in embedded systems. Thus, we do not consider physical attacks such as side-channel attacks (SCA) as these attack types are beyond the scope of this study.

### 3.2. Implementation Challenges

DEMIX targets very lightweight embedded systems commonly used in IoT environments. These systems usually comprise a single core, a few peripherals, and flash and SRAM memory and are often utilized as endpoint sensors or simple actuators. Given these systems’ limited resources, several challenges here refers to the characteristics essential to be achieved when implementing DEMIX, denoted as C1, C2, C3, and C4 as follows.

**C1.** **Efficient Domain Description:** In contrast to larger systems that support a memory management unit (MMU) for virtual memory support, these systems typically use MPU for physical memory protection. MPUs usually have a minimal configuration entity, making it challenging to define multiple domains efficiently with limited resources.**C2.** **Efficient Domain Protection:** The increased number of unprivileged domains delivers a complex challenge in managing access permissions. The limited resources of embedded processors make configuring permissions for multiple domains difficult. For example, cross-domain entry protection, which traditional techniques support using entity tables, can become inefficient as the number of domains increases**C3.** **Reduce Switching Overhead:** Minimizing the overhead of domain switching is another challenge. Cross-domain function calls typically require a significant overhead regarding processing time and memory usage. The challenge is to minimize this overhead while maintaining the security of the domain switch process.**C4.** **Minimum Hardware Overhead:** Balancing security and hardware overhead is a major challenge in implementing embedded systems. The use of multiple unprivileged domains can improve security, but it also results in increased overhead. The challenge is to find the optimal balance between security and hardware overhead to meet the requirements of embedded systems with limited resources.

### 3.3. Addressing Threats and Challenges

After discussing and analyzing the threat model and challenges, we determine the following features of DEMIX to address them:
**Hardware-Based Domain Identifier:** This feature ensures that each domain is uniquely identified and isolated from other parts, providing a secure boundary for each domain. This identifier prevents T3 (i.e., Tampering with Domain Switching Process) as it is managed through hardware inside the processor. At the same time, it also counteracts C3 (reduced switching overhead) because it is easy to switch which domain the processor is performing by changing the identifier.
**Identifier-Based Memory Authorization:** This feature provides memory protection based on the domain identifier, allowing only authorized domains to access specific memory regions. Moreover, it ensures that each domain only accesses a legitimate memory area in response to T1 (unauthorized access to sensitive memory). Furthermore, since this method can be implemented using the MPU instead of the MMU, it is well suited for low-end systems, where memory resources are limited and hardware overhead is a concern. As a result, it helps address two design challenges, C1 (Efficient Domain Description) and C4 (Minimum Hardware Overhead), for lightweight embedded systems.
**Instruction-Level Domain Authentication:** This feature validates the domain identity using instructions, providing secure access to sensitive functions. This prevents T2 (i.e., Malicious Cross-domain Function Calls) by securing domain access, which enforces the rule that access to this region must perform authentication. Additionally, this method eliminates the need for a domain entity table required by traditional technologies and enables efficient and fine-grained control over which domains can access specific functions. Instruction-level domain authentication can address C1 (Efficient Domain Description), C2 (Efficient Domain Protection), and C4 (Minimum Hardware Overhead), whereas securing cross-domain function calls in lightweight systems.
**Instruction-Level Secure Memory Access:** This feature enables secure memory access at the instruction level, protecting sensitive data. It ensures that only authorized instructions can access sensitive data while protecting them from accessing shared memory, thereby addressing the threat of T4 (Exploiting Shared Memory). It also prevents domain fragmentation by reducing the need for separate domains or privilege levels to protect sensitive data, thus addressing C1 (Efficient Domain Description) and C2 (Efficient Domain Protection).

## 4. DEMIX Proposal

This section presents a detailed description of DEMIX. First, we provide the design overview of DEMIX, then discuss the main elements of DEMIX: the *Domain-Enforced Memory Isolation* in Section 4.2 and *the instruction-level domain isolation* in Section 4.3. Subsequently, we also provide the illustration of the memory protection functionality, which ensures access to sensitive data and functions is restricted to only authorized domains, in Section 4.4. Finally, we elaborate on how domain isolation functionality is achieved in Section 4.5.

### 4.1. Design Overview

Our proposed solution, DEMIX, presents a novel and efficient approach to memory isolation designed to meet resource-constrained IoT devices’ security needs. The solution provides these capabilities through two fundamental techniques: *Domain-Enforced Memory Isolation* and *instruction-level domain isolation*.

*Domain-Enforced Memory Isolation* supports multiple unprivileged domains’ inter-domain isolation by utilizing a hardware-based domain identifier and identifier-based memory authorization. This technique provides an efficient domain memory isolation method while eliminating the switching overhead required. We implement this by integrating a Domain Protection Unit (DPU), which assigns domain owner, trust, and protection levels to each region of system memory. The DPU, working with the processor’s current domain state, handles access to the corresponding address space.

*Instruction-level domain isolation* enhances the security of each domain and enables fine-grained control over access to specific regions. This technique provides secure access to sensitive functions and data regions through instruction-level domain authentication and instruction-level secure memory access. These features help ensure intra-domain isolation, providing added security to each domain. We implement the DTI instruction for sensitive functions and the Secure Load/Store instruction for data.

### 4.2. Domain-Enforced Memory Isolation

The unprivileged domain overcomes the constraints of traditional software-defined memory isolation approaches by implementing vertical partitioning. Vertical partitioning serves within the unprivileged level, whereas horizontal partitioning is based on the privilege level. We support this by expressing, through an identifier, which domain is running in the processor and authorizing memory access through that identifier. Based on the character of this mechanism, we call this technique Domain-Enforced Memory Isolation. The implementation of Domain-Enforced Memory Isolation is achieved through two components: the domain state and the domain policy. The domain state is supplied by the processor’s Domain State Register (DSR), whereas the Domain Protection Unit (DPU) manages the domain policy.

#### 4.2.1. Domain State Register (DSR)

DEMIX serves two Control and Status Registers (CSR) to handle the processor’s domain state: the processor’s Domain State Register (DSR) and the Previous Domain State Register (PDSR). The DSR represents the currently active domain in unprivileged mode. It expresses either the code’s domain passed to the pipeline before or the initialized domain in privilege mode. The DSR is utilized for two primary purposes in DEMIX. Firstly, it acts as an identifier to restrict the execution of the processor within a specified domain, allowing the domain policy with multiple domains. Secondly, the DSR allows the processor to monitor domain changes by tracking the executing code’s domain and performing domain-switching validation based on protection features. Although the processor’s DSR provides partitioning between domains in unprivileged mode, the PDSR provides communication between privilege levels. When switching between privileged modes, the PDSR backs up the DSR or restores the respective domain values. Specifically, switching to privileged mode saves the DSR to the PDSR, and switching to unprivileged mode restores the DSR to the PDSR. This mechanism ensures the protection and preservation of the DSR while enabling smooth transitions between privilege levels.

#### 4.2.2. Domain Protection Unit (DPU)

The Domain Protection Unit (DPU) configures address spaces and protects memory access. As an extension of the Memory Protection Unit, the DPU implements a domain policy for each memory region, allowing for the simultaneous management of address spaces for multiple domains. The domain policy assigns ownership, trust, and protection levels to each memory segment, which determines the identifier for the memory segment (ownership), the authorized domain states for accessing the memory segment (trust), and the level of security for the memory segment (protection level). In domain policy, ownership and trust are the essential resources that control the allocation and protection of the domain’s memory. Domain ownership serves as the identifier for the memory segment and allocates the processor’s DSR during execution. On the other hand, the trust activates the memory segment based on the DSR, permitting access to only those domains that have been granted authorization. The sequence of domain identifiers through the DSR ensures that when executing code with specific ownership, the processor can only access memory segments whose ownership it trusts.

#### 4.2.3. Inter-Domain Isolation

Inter-domain isolation refers to the ability to separate address space between different domains. This feature is essential to protect sensitive memory in one domain from unauthorized access in the untrusted domain. Figure 2 illustrates our proposed architecture for Domain-Enforced Memory Isolation, i.e., DEMIX, with the operation described as follows. During memory access, the processor provides the processor’s DSR to the DPU. Based on this DSR information, the DPU calculates the set of memory segments that the processor is authorized to access and provides the corresponding address space. The DSR acts as an identifier for reconstructing the processor’s accessible address space. In the figure, the domain switching process and memory protection mechanism in DEMIX are presented. For switching to User Domain1, the privilege mode sets the PDSR to Domain1 and transitions to the code in User Domain1. Upon transitioning from privileged to unprivileged, the value in the PDSR is copied to the DSR, effectively pointing to Domain1. As a result, the processor can only access memory segments authorized for Domain1.

The ownership of a policy is associated with the domain that will occupy the processor’s DSR when the code segment is executed. As previously shown in Figure 2, the code in Domain3 is accessible from Domain1 as specified by the trust configuration. When Domain1 accesses the code segment of Domain3, the DSR switches to Domain3, reconfiguring the address space accessible to Domain3. This mechanism enables fast domain switching, as the domain ownership enforces the address space. However, if the access is from an untrusted domain, the access will be blocked, and an exception will be thrown. For example, in the figure, the code segment in Domain2 does not trust Domain1. Therefore, the code in Domain2 cannot be executed when the processor is in the state of Domain1.

### 4.3. Instruction-Level Domain Isolation

Intra-domain isolation enhances the protection of sensitive code and data by enabling granular access control within a single domain. This approach offers differentiated access privileges based on the functionality within a single domain, which serves as an additional layer of protection. Unfortunately, most memory isolation techniques do not consider intra-domain isolation, making constructing configuration challenging and inefficient as the number of domains increases in a multi-domain system. To address this issue, DEMIX introduces an instruction-level domain isolation technique that eliminates these inefficiencies and provides robust protection. DEMIX provides extended instructions for intra-domain isolation of code and data: the Domain Target Identifier (DTI) and Secure Load/Store. The DTI instruction authorizes access by individual tasks within a domain, whereas the Secure Load/Store instructions ensure safe access to secure data segments.

#### 4.3.1. Domain Target Identifier (DTI)

To ensure the security of cross-domain function calls, DEMIX implements the Domain Target Identifier (DTI) instruction, which provides two forms of safeguarding. First, it specifies the accessible domains based on the functionality of the tasks, enabling fine-grained access control within a multi-domain system. Second, it guarantees the entry address of the domain task, thereby mitigating the possibility of unstable behavior resulting from malicious code. As previously described in Section 4.2.2, the domain policy enables the configuration of trusted domains that can access the memory segment. On the other hand, the DTI instruction provides an additional layer of security by designating accessible entrances by domains. The protection level of the domain policy can establish this additional security layer. If the memory segment’s protection level is set to normal, trust verification based on the DSR is conducted. In contrast, if set to secure, the DTI verification is also performed, in addition, to trust verification.

The DTI verification process consists of two stages. In the first stage, the processor verifies whether the loaded instruction is a DTI instruction when the processor’s DSR and the instruction’s ownership differ. This stage serves as an access point verification when branching to a secure code in another domain, guaranteeing the function execution entrance. In the second stage, the processor’s DSR is verified using DTI instruction. The DTI instruction encodes the trust set as an immediate value. If the processor’s DSR is within the DTI’s trust set, access is allowed. As a result, each task in secure code can specify which domains can access it based on the functionality.

#### 4.3.2. Secure Load/Store

DEMIX supports distinct load/store instructions to enable intra-domain isolation for data segments within a domain, including normal and secure load/store instructions. Each data memory segment can specify the type of load/store instruction needed to access its data using the protection level. The processor must access each data segment with the appropriate load/store instruction. Note that the normal load/store instructions are designated for general access, whereas the secure load/store instructions are designated for secure access. Differentiating secure access to data segments can mitigate the risk of malicious or erroneous code accessing sensitive data within a domain. For example, in the case of a shadow stack, the processor should only allow access to that memory from valid locations in the code. Secure load/store instructions can restrict the scope of these access regions on an instruction level, thereby enhancing security.

#### 4.3.3. Intra-Domain Isolation

Instruction-level domain isolation technology offers a more fine-grained approach to access control for code and data, allowing DEMIX to implement intra-domain isolation and assign varying permissions to resources within a single domain. Furthermore, as permissions are bonded to instructions, DEMIX eliminates the additional MPU configuration, such as entity tables, which were required in previous research. Instead, DEMIX distributes these permissions within the code, offering more efficiency in securing multiple domains with fewer entities.

### 4.4. Memory Protection Functionality

The protection strategy in DEMIX is based on the domain policy, which assigns an access protection strategy to each memory segment. The policy ensures that access to memory segments is only granted to authorized domains and provides a more elevated level of security in the system. The protection strategy is mainly differentiated by whether a memory segment is code or data and whether its protection level is normal or secure. The protection level supports granting access to highly distributed memory regions through instruction-level domain isolation techniques, which enables the creation of fine-grained, highly distributed permissions within the instruction unit. Figure 3 illustrates the memory protection functionality in DEMIX. As shown, each segment in the DEMIX’s address space can be shared among different domains through trust configuration or have extra privileges granted through intra-domain isolation at the instruction level. This section provides more information on the memory protection functionality of each segment under these configurations.

#### 4.4.1. Access to Normal Code Segment (NCS)

The Normal Code Segment (NCS) is designated with a normal protection level for executable code, providing a basic level of protection through inter-domain isolation. This segment protects against executing internal functions in untrusted domains using the domain policy’s trust entry. If accessed from an untrusted domain, an exception is raised.

#### 4.4.2. Access to Secure Code Segment (SCS)

The Secure Code Segment (SCS) is designated with a secure protection level for executable code, providing both inter-domain and intra-domain isolation. Access to the SCS, even for trusted domains, is limited to specific addresses through the DTI instruction. At the same time, the DTI instruction encodes the trusted domain which can execute, so access to that location depends on the processor’s current execution domain. This enables the SCS to define decentralized entry points for each domain within a single segment, addressing a limitation of traditional memory isolation techniques that struggle to define highly distributed regions.

#### 4.4.3. Access to Normal Data Segment (NDS)

The Normal Data Segment (NDS) is specified with a normal protection level for data. This segment provides instruction-level isolation where only normal load/store is accessible but categorized it as providing only inter-domain isolation based on the general purpose usage of those instructions. Access to this segment is allowed when the processor executes the trusted domain’s normal load/store instructions. An exception is raised if the processor executes either within the untrusted domain or the secure load/store instructions.

#### 4.4.4. Access to Secure Data Segment (SDS)

The Secure Data Segment (SDS) is designated with a secure protection level for data and is intended for specialized purposes within the domain. Access to the SDS is only allowed through secure load/store instructions. Attempts to access the SDS using normal load/store instructions or from an untrusted domain will raise an exception. The use of secure instructions, as opposed to normal instructions used for general purposes, allows for the distribution of accessible memory regions for the SDS throughout the code at the instruction level, resulting in finer control over access to sensitive information and improved security.

### 4.5. Domain Isolation Functionality

In memory protection, a domain is a logical division of memory space with assigned separate access regions and authority for each partition. Each partition acts as a unit to protect the system’s memory by organizing a specific set of memory regions that the processor can access. DEMIX, as proposed, supports multiple domains, each isolated at the unprivileged level. During execution, the processor applies the appropriate protection and occupancy process based on the code’s ownership. The DPU uses this ownership to activate memory segments to provide a legitimate address space. This mechanism does not only isolate sensitive memory within a domain but also enables efficient domain switching.

#### 4.5.1. Domain Protection

To ensure the security and privacy of each domain, it is necessary to implement domain protection by restricting access to sensitive data and functions to authorized domains only. DEMIX provides this protection and sharing of memory segments owned by each part through Domain-Enforced Memory Isolation technology. Figure 4 illustrates this domain protection functionality, with the example for the case of Domain1.

To access the code in Domain1, the processor must validate the current domain state with the DPU. The DPU only allows access to segments that it trusts based on the current domain state of the processor. In the figure, Region A is accessible when the processor is in Domain2 and Domain3, but Region B only allows access to Domain3. Additionally, Region A is a secure segment (i.e., SCS), a code segment of DEMIX that is protected by intra-domain isolation. The non-owner domain must perform DTI verification when accessing this segment. Therefore, in the figure, the function *foo* in Region A can be called from Domain2 and Domain3, but the function *bar* is only accessible from Domain3. In other words, function *foo* is public to Domain2 and Domain3, whereas function *bar* is protected and can only be accessed by Domain3.

If the cross-domain function call is validated, the domain state is set to Domain1, and the processor can access Regions C and D accordingly. Both regions are data segments that only Domain1 trusts. As a result, Regions C and D are inaccessible to other domains and can only access them after legitimate cross-domain function calls to Domain1. This ensures that data within a domain is protected from unauthorized access and manipulation. On the other hand, Region E is owned by Domain2 and has trusts both Domain1 and Domain2. As a result, when switching from Domain2 to Domain1, it remains accessible. This illustrates how Domain-Enforced Memory Isolation allows for the controlled sharing of data between trusted domains while restricting access to unauthorized domains.

Finally, Regions C and D are configured with different protection levels: secure (i.e., SDS) and normal (i.e., NDS). The processor can only access each part through the dedicated load/store instruction. This ensures that Domain1’s sensitive data is only accessible from specific locations while preventing any possibility of Domain1’s sensitive data being leaked to other domains’ shared segments, such as Region E.

#### 4.5.2. Domain Switching

To execute a specific task in a trusted domain, the processor must change the accessible memory region for that domain before running that task. This process is known as domain switching and it is designed to be highly efficient with minimal overhead. By updating the DSR provided by the processor, DEMIX can instantly reconfigure the accessible memory region for that domain without negatively impacting the system’s performance. When accessing a task in the NCS region, the processor changes the DSR during the instruction fetch. As a result, there is no overhead compared to a typical call instruction. DEMIX also provides a DTI instruction specifying accessible domains while ensuring task entrance. It further restricts cross-domain access to authorized task entries within the NCS. In this region, if the validation of the cross-function call is successful, the processor will update the DSR with the ownership and apply the new address space. This operation incurs an overhead of only one cycle. These overheads are more efficient than previous studies that aim to guarantee cross-domain function call entry and regulate accessible memory regions.

## 5. Evaluation

### 5.1. Hardware Implementation

To verify and measure the hardware overhead of our technique, we first develop a prototype that supports DEMIX technology using the RISC-V Ibex processor. In this section, we describe the hardware implementation of applying DEMIX technology on the RISC-V processor.

#### 5.1.1. DPU Implementation

As described in Section 2.1, the RISC-V architecture already provides Physical Memory Protection (PMP) for memory isolation. Although PMP offers software-defined memory protection, it only supports fixed address spaces, unlike the proposed DEMIX. To implement the memory protection mechanism of DEMIX, we have extended the Configuration and Status Register (CSR) by adding the *dpucfg* to provide domain policy to the PMP Entity. The *dpucfg* includes the domain policy of ownership, trust, and protection level for each memory segment defined within the RISC-V architecture. The ownership of a domain can be represented through binary encoding, whereas the protection level can be fixed to a single bit per entry. The trusted domains are defined through bit encoding for each domain ownership, with the number of bits required equal to the number of domain ownership supported by the processor. For example, supporting 4 domains requires 7 extra bits of register per entry, whereas 21 additional bits are required for 16. Larger cases are not considered as they exceed the maximum number of entries supported by the RISC-V PMP.

#### 5.1.2. Instruction Set Extension Implementation

We have enhanced the RISC-V instruction set architecture (ISA) by extending dedicated instructions for memory access. Using the custom instruction encoding capabilities of the RISC-V ISA, we have implemented DTI instructions to enforce cross-domain function call entry and Secure Load/Store instructions to restrict access to secure data. The DTI instruction restricts memory access during cross-domain function calls from a shared domain. The instruction encodes the accessible domain state as an immediate value using bit encoding, determined at compile time. Separate permissions can be configured for different domain ownership, allowing for fine-grained control over memory access. For example, the DTI instruction with an immediate value of 0x03 permits access to the memory addresses when the domain state is 0 and 1. In contrast, the DTI instruction with an immediate value of 0x02 grants access only when the domain state is 1. The extended DTI instruction prototype in our implementation supports a maximum of 16 domain states, which aligns with the maximum number of PMP entries in the RISC-V architecture. The secure load/store instructions provide access to data regions with secure protection levels, and they are designed to ensure that authorized locations can only access sensitive data. These instructions are replicated from the load/store instructions in RISC-V but with a different encoding location in the ISA. If the load/store is executed at the replicated encoding, the DPU allows access to the secure data segment. For example, the lw in the original RISC-V architecture is an instruction to load word data from the normal data segment (NDS) into a register file. In contrast, the seclw instruction is the corresponding secure load/store instruction that loads word data from the secure segment (i.e., SDS) into a register file.

#### 5.1.3. DSR Implementation

DSR is located in the processor’s instruction fetch pipeline. During instruction fetch, the processor manages the DSR according to the domain policy of the loaded instruction. When the code’s domain ownership and DSR differ, the processor performs domain validation. If the verification is successful, the DSR will change to the newly loaded ownership. For DTI validation, validated instructions are masked as NOP and passed to the processor pipeline to avoid performing unnecessary actions. Additionally, the Domain State is a value that changes according to the domain policy in the user privilege level, but a direct modification must be prohibited. At the same time, it must be returned after performing other code, such as Interrupt Handler. To support this, we extend PDSR in the CSR. When the privilege level is changed from the machine to the user due to the MRET instruction, the DSR value is copied to the PDSR. Furthermore, if there is a change from user to machine privilege due to the ECALL or Interrupt, the DSR value is stored in the PDSR.

### 5.2. Hardware Evaluation

The RISC-V Ibex Core was selected for our prototype implementation, with the processor configuration set to *rv32imbc*. The Ibex Core was chosen because it is a very lightweight RISC-V processor, which makes the hardware overhead of extending DEMIX more visible. To accurately measure the overhead caused by the DEMIX extension, additional features such as cache and branch prediction were disabled. We tested designs that support 4, 8, and 16 user-domain partitions on the Ibex Core with 16 physical memory protection regions. Although 16 user domains may be excessive for typical use cases, they were included in our tests to measure the hardware cost of the DEMIX solution. For testing, we implemented our technique on a Xilinx Artix-7 35T FPGA, with the hardware cost results for our configuration as shown in Table 1.

As shown in Table 1, the increase in Look-Up Table (LUT) resource usage as the number of multiple domains increases is relatively insignificant. We measured a slight 2.8% and 3.4% cost increase for 4 and 8 user domain partitions, respectively. Furthermore, we successfully implemented our solution with only 5.1% overhead, even when supporting 16 user domains, the maximum number of regions PMP supports. On the other hand, Flip-Flop incurs a more significant increase in cost due to its need for CSR to define the domain policy. Furthermore, we measured only a 4.8% and 7.9% cost increase for 4 and 8 user-domain partitions, respectively. Furthermore, even with 16 user domains implemented for experimental, we counted an overhead of 13.6%. It is important to note that the Ibex processor is a very lightweight RISC-V processor, and thus the resource overhead metrics should reflect this fact. We created an environment to evaluate the maximum overhead of extending DEMIX and measured the results accordingly. Despite these constraints, the maximum measured overhead is below 14%, and partitions for 4 to 8 user domains can be implemented with less than 8% overhead.

### 5.3. Software Evaluation

Figure 5 illustrates the overhead required for domain switching. For software evaluation, the most essential overhead lies in domain switching. Part A of the figure illustrates the use of privileged software to perform a domain switch. This is conducted by switching to the machine privilege level via *ecall* instruction and then performing a domain switching via privilege software. To compare domain-switching overhead, we developed privileged software that performs the function of DEMIX. The software was developed as an elementary function that verifies the legitimacy of the old domain, changes the PMP configuration to limit the memory area the new domain can access, and then switches to that function. Even excluding other user-trap functions, it took 76 cycles to perform a domain switch. Part B of the figure illustrates the overhead of switching domain supported by DEMIX. DEMIX supports immediate address space conversion through Domain-Enforced Memory Isolation while supporting domain verification through DTI instructions. DEMIX supports immediate address space conversion through Domain-Enforced Memory Isolation while supporting verification through DTI instructions. As a result, domain switching using existing privilege software can be performed in a single instruction. The software overhead may vary depending on the features supported, but using DTI instructions incurs only one cycle of overhead compared to the 76 cycles of overhead developed for comparison.

DEMIX reduces the overhead of switching domains by minimizing the process, which is particularly advantageous when the amount of executed code is small after switching. The *fbctl* function in the BEEPS Benchmark takes 8893 cycles when executed via privilege-based domain switching but 8710 cycles when using DEMIX. However, for functions with short internal execution, such as *crc32*, the reduced overhead using DEMIX is significant at 98 cycles compared to 287 cycles when executed via privilege-based domain switching. Note that due to the separation of the timer domain and the benchmark execution domain, a round-trip switching overhead is added. Figure 6 presents the roundtrip switching overhead per code size. In other words, it compares the execution time of DEMIX with switching via privileged software as a function of the code size being executed relative to a normal function call. As shown, DEMIX incurs nearly the same execution time irrespective of the code size, whereas the overhead increases with the number of smaller code sizes when privileged software is involved.

The comparison of overhead for several related techniques is presented in Table 2. In detail, for memory isolation with MMU [27,30], utilized in most unprivileged domain research, page tables can be leveraged to support more unprivileged domains. However, using MMU for memory isolation is unsuitable for lightweight IoT environments as it requires significant resources. Additionally, managing more domains with MMU requires software intervention in the switching process, which can cause overhead. On the other hand, leveraging MPU for memory isolation allows for fast domain switching as it supports fewer memory management features than MMU. This makes it ideal for systems with real-time requirements or limited resources. Nevertheless, there are limitations to defining multiple domains using a limited number of MPU entries, which forces most research to provide only a very limited number of domains and trade it off using another approach such as by using privileged software to manage domains with single unprivileged domain [34] or using only a secure part to protect essential resources [21].

To overcome this limitation, DEMIX incorporates Domain-Enforced Memory Isolation and instruction-level memory isolation techniques that facilitate efficient domain definition and switching across multiple domains. As a result, DEMIX can switch domains in 0 to 1 cycle, compared to the 120 to 160 cycle overhead of Multizone [34], which relies on privileged software for domain switching. Additionally, unlike TrustZone-M [21], DEMIX can scale for a larger number of domains. The 16 MPU entities can utilize software-defined domain policies to partition unprivileged domains. Simultaneously, by supporting domain-specific permissions and entrance positioning at the instruction level, DEMIX eliminates the need for a separate MPU entity and additional indirect jump, as required in the entities table approach [19,21]. This shows that our approach to memory isolation by domain offers a scalable solution for embedded systems. In particular, by efficiently utilizing a limited number of MPU entities, DEMIX can support more domains without sacrificing performance. In conclusion, our approach to memory isolation by domain provides a novel solution for enabling the efficient and secure execution of multiple domains on a single processor in embedded systems. It enables efficient domain definition and switching across a larger number of domains using a limited number of MPU entities. This scalability is achieved without compromising performance, making our approach well-suited for embedded environments that require a real-time response or where domain switching frequently occurs.

## 6. Discussion

In this section, we discuss the assessment of our approach and elaborate on its comparison with existing techniques.

### 6.1. DEMIX Functionality Assessment


**Flexible domain configuration:** DEMIX provides flexible domain configuration by implementing the policy on the Domain Protection Unit (DPU). The domain policy is protected by a privilege level and can be configured through software. This allows the system to reconfigure the domain through a domain handler protected by the privilege level, providing the flexibility to adapt the domain configuration to the system’s requirements.
**Unprivileged domain isolation:** DEMIX provides unprivileged domain isolation by implementing inter-domain isolation technology. This technique is implemented through domain states and policies that assign a list of accessible domain states to each memory segment. This allows the accessible segments to be adjusted by changing the domain state, providing unprivileged partition by domain state.
**Multiple domain partition:** DEMIX supports multiple domain partitioning by using domain states as identifiers to restrict memory access for each segment. The number of domains supported by DEMIX is statically limited by hardware support, but it still provides more user domains than existing technologies. This solution is reasonably suited for lightweight embedded systems that typically have a limited number of memory configuration entities. The intra-domain isolation technology in DEMIX enables fine-tuned access control for memory segments, allowing for the efficient representation of multiple domains with limited resources.
**Lightweight domain switching:** DEMIX enables lightweight domain switching by eliminating the need for a secure monitor or privilege mode during cross-domain function calls. Instead of relying on a secure monitor, it controls the domain state of the processor through predefined code ownership, resulting in a lightweight and streamlined switching process. As a result, DEMIX enables more rapid responsiveness in embedded systems compared to traditional approaches. Additionally, it requires at most one cycle overhead compared to a typical function call, so it can adapt to more granular domain partitions with frequent switching.
**Task-specific authorization:** DEMIX implements task-specific authorization by providing fine-grained access control at the instruction level. This is achieved through the use of DTI instructions, which restrict access to each unit task in a code segment to only trusted domains. This approach allows highly distributed entry points to be defined for each domain within a single memory segment, resulting in more granular access control to sensitive functions. Compared to traditional techniques that use entity tables or secure monitors, DEMIX’s approach supports multiple domains with fewer memory configuration entities and reduced switching overhead.
**Secure data protection:** DEMIX provides secure data protection through secure data segments (SDS) and secure load/store instructions. The secure load/store instructions provide access to the SDS and are designed to ensure that only authorized locations can access sensitive data. This feature allows fine-tuning access to sensitive data that needs to be separated and protected, providing an additional layer of security within a single domain.


### 6.2. Comparison with Existing Techniques

DEMIX is a novel memory protection technology that provides a more flexible and efficient solution for protecting sensitive data than existing technologies. It provides fine-grained access control over the embedded system, defining distinct access privileges based on functionality and sensitivity. This ensures that only authorized domains can access sensitive code and data, reducing the risk of security threats and improving overall system performance. Table 3 compares DEMIX and existing techniques, with Memory Protection Key (MPK) excluded from the comparison as it is outside the scope of the lightweight embedded system. Nevertheless, it is noteworthy that the Domain-Enforced Memory Isolation technique provided by DEMIX can be considered a simplified version of MPK, merging the access permission settings for each virtual page in MPK to a specific code segment, whereas DEMIX does not provide the scalability of MPK, which allows access rights to be modified for each page in user space software, it eliminates the security vulnerabilities associated with this capability. Given that our implementation targets lightweight embedded systems, this trade-off is acceptable, as it provides a secure solution within the constraints of these systems.

DEMIX’s intra-domain isolation offers an advantage over existing techniques by providing fine-grained access control at the instruction level. This allows highly distributed entry points to be defined for each domain within a single memory segment, resulting in more granular access control to sensitive functions. This approach supports multiple domains with fewer configuration entities and reduced switching overhead compared to traditional techniques using entity tables or secure monitors. In addition, the secure data protection provided by DEMIX’s secure data segments and secure load/store instructions give an additional layer of security within a single domain, ensuring that only authorized locations can access sensitive data.

Lastly, to clarify the difference between DEMIX from our original version (i.e., RIMI), we provide the discussion as follows. In RIMI [22], we also proposed a memory access decentralization technique that can offer fine-grained access control at the instruction level. However, it only supports two domains, and authorized validation is solely based on instructions. As a result, this method’s scalability was limited since defining multiple domains required adding as many instruction sets as domains to be supported. Additionally, for implementation, RIMI was only emulated through Spike instead of a hardware platform (e.g., FPGA). On the other hand, DEMIX is built upon the idea of instruction-level authorization, but now provides a more generalized definition of the domain, whereas we borrow some concepts from RIMI (e.g., the secure load/store), most of the work is new. In essence, we introduce a Domain-Enforced Memory Isolation technique that efficiently manages and defines multiple domains. This technique provides a user-level partitioning of multiple domains on an embedded processor without requiring privilege-level intervention. Each domain is configured to efficiently serve its address space by functioning as the new identifier, whereas the instruction-level isolation technology from RIMI strengthens the security of each domain —thus allowing for additional layers of data —enhanced features like configuring permissions within a domain or per task are not present in RIMI. Lastly, DEMIX was developed and evaluated on an FPGA instead of only on an emulator. As a result, we not only achieve user-level partitioning of multiple domains on an embedded processor but also fine-grained access control within a domain compared to our previous version, which is of significant advantage.

### 6.3. Limitations and Future Work

This paper presented a novel methodology for supporting multiple domains in lightweight embedded processors, and we have applied it to RISC-V processors. We provide an efficient solution for multi-domain isolation in these environments. However, although we offer significant advantages for these applications, it has several limitations when applied to more general-purpose systems. One of the primary limitations of DEMIX is the static limited number of domains it can support due to hardware constraints. In particular, the number of partitions supported by DTI instructions is limited because of bit width constraints. In addition, although our proposed technique is efficient for static domain configuration in a lightweight environment, it is unsuitable for dynamic domain adaptation. Domains in our proposal are statically specified at compile time, which is practical for splitting a single domain into subdomains to increase security or provide fixed multi-domain isolation. However, merging subdomains dynamically or changing domain ownership is difficult, which limits the to adapt dynamic environments.

In the future, we aim to further demonstrate the practicality of the DEMIX solution by deploying a trusted execution environment and evaluating its effectiveness in providing security solutions and protection against cyber-attacks. In addition, we will find methods for dynamically merging domains or changing domain ownership. It would make the solution more flexible for dynamic conditions, allowing it to adjust to changing requirements and provide more comprehensive protection against security threats.

## 7. Conclusions

In this paper, we presented DEMIX, a novel memory isolation technique suitable for lightweight embedded systems, driven by the motivation to provide a security solution to guard against their vulnerability to malicious software. Our proposal offers a flexible configuration for multiple domains and efficient domain switching through two elements: Domain-Enforced Memory Isolation and instruction-level domain isolation, which thoroughly validates a domain’s access to sensitive tasks and provides secure access to sensitive data. This combination of flexible domain configuration and secure protection makes our solution well-suited for delivering isolated peripheral control software with secure call gates, thus offering security for the increasing protection needs of lightweight embedded systems. To evaluate the performance of our technique, we implemented our prototype on a RISC-V Ibex system, evaluating the effectiveness and overhead of our proposed solution. Our software evaluations have demonstrated that the overhead of switching to smaller code execution is negligible. Given that the code size for controlling sensors inside the SoC or external I/O in an IoT environment is very small, our technique can be safely decoupled for each peripheral without any additional overhead. As verified on the FPGA system, the overhead shows that our solution can be adaptable to the lightweight embedded system by only taking 8% of hardware overhead on an eight-user domain implementation in Ibex Core, a lightweight RISC-V processor.

## Figures and Tables

**Figure 1 sensors-23-03568-f001:**
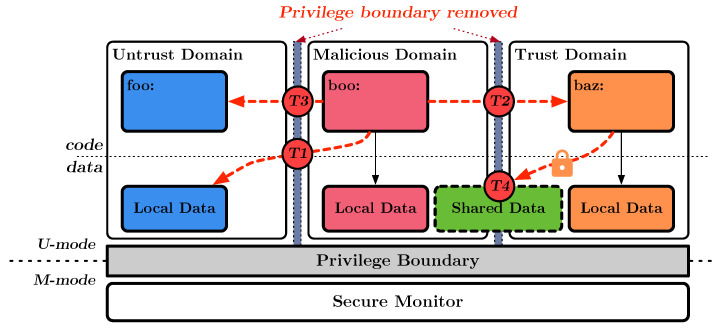
Threats in unprivileged domain.

**Figure 2 sensors-23-03568-f002:**
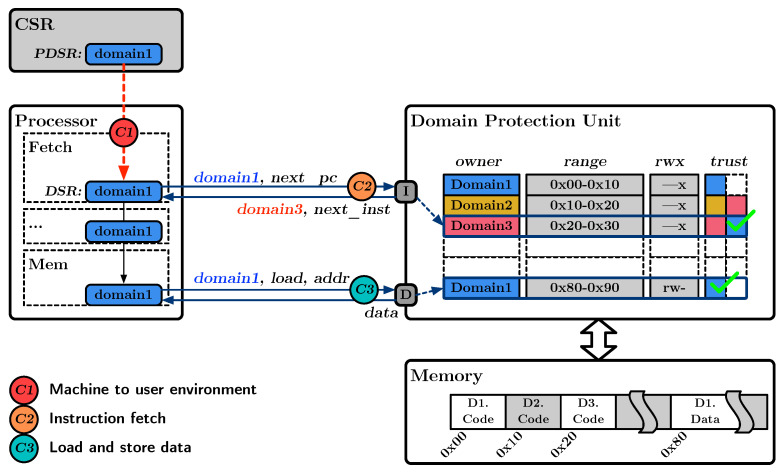
Proposed architecture of Domain-Enforced Memory Isolation.

**Figure 3 sensors-23-03568-f003:**
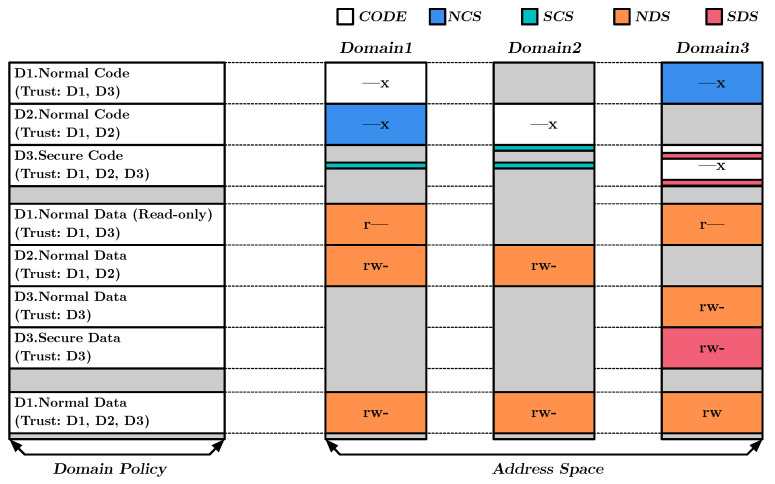
Memory protection functionality in DEMIX.

**Figure 4 sensors-23-03568-f004:**
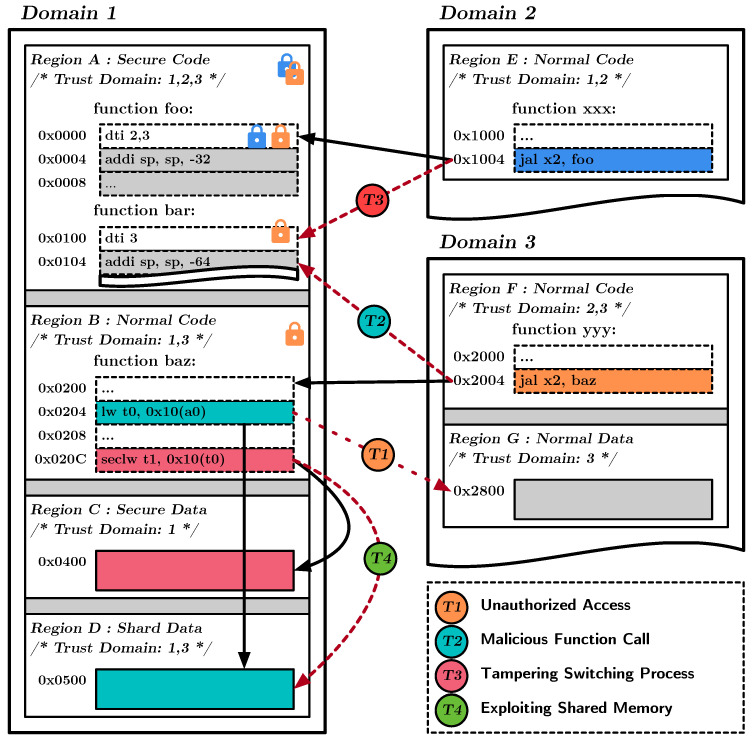
Domain protection functionality in DEMIX.

**Figure 5 sensors-23-03568-f005:**
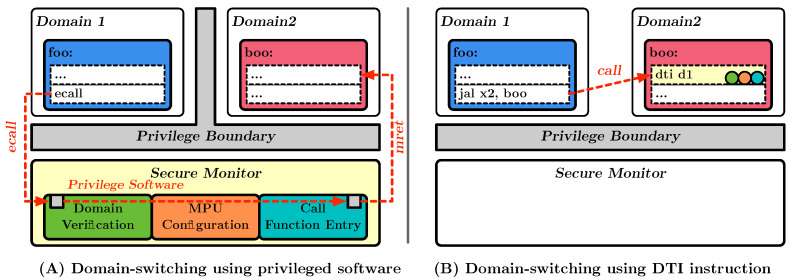
Domain-switching overhead in DEMIX.

**Figure 6 sensors-23-03568-f006:**
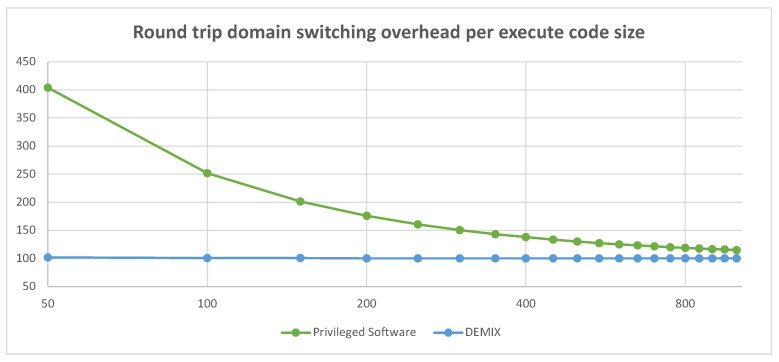
Roundtrip-switching overhead per code size.

**Table 1 sensors-23-03568-t001:** FPGA implementation result. The symbol (“+”) signifies the increase in resources in percentage compared to the standard (i.e., no security) implementation in Ibex.

Resource Type	Ibex Original	Ibex + DEMIX (Overhead from Original)		
		4 Domains	8 Domains	16 Domains
LUT	8277	8512 (+2.8%)	8559 (+3.4%)	8701 (+5.1%)
FF	2545	2666 (+4.8%)	2746 (+7.9%)	2891 (+13.6%)
DSP	1	1	1	1

**Table 2 sensors-23-03568-t002:** Domain-switching overhead comparison.

	ERIM [27]	DONKY [30]	Trustlite [19]	TrustZone-M [21]	Multizone [34]	DEMIX
**Processor**	Intel	RISC-V	Siskiyou Peak	ARM	RISC-V	RISC-V
**Protection Module**	MMU	MMU	EA-MPU	MPU	MPU	MPU
**Unprivileged Domain**	16	1024	∼16 *	2	1	∼16 *
**Switching Overhead**	60–99 cycles	160 cycles	A few cycles	A few cycles	120–160 cycles	0–1 cycle
**Domain verification**	Software	Software	Entries tables	Entries table	Software	Instruction

* Note: The number of unprivileged domains supported by Tustlite and DEMIX can vary depending on the domain configuration. Sixteen is the number of MPU entities. If one domain requires four entities, then the number of unprivileged domains that can be supported is four.

**Table 3 sensors-23-03568-t003:** Detailed functionality comparison with existing MPU-based techniques.

Functionalities	EA-MPU	TrustZone-M	RISC-V PMP	RIMI	DEMIX
Flexible domain configuration	**High**	Low	**High**	**High**	**High**
Unprivileged domain isolation	**Supported**	**Supported**	Not Supported	Limited	**Supported**
Multiple domain partition	**N**	2	N/A	2	**N**
Lightweight domain switching	Medium	Medium	N/A	Limited	**High**
Task-specific authorization	Entries tables	Entries tables	N/A	Not Supported	**Instruction**
Secure data protection	Limited	Not Supported	N/A	**Instruction**	**Instruction**

## Data Availability

Data are available from the authors on request.

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
