# Peer review of "DEMIX: Domain-Enforced Memory Isolation for Embedded System"

_sensors, 2023, doi:10.3390/s23073568_

Round 1

Reviewer 1 Report

This article addresses the issues of security in resource-constraint embedded devices (IoT applications).  To do this, authors have proposed a domain-enforced memory isolation technique.  It has been claimed that the proposal provides useful user-domain control in terms of efficiency, safety, switching time, and security.  Overall, the article is well-written and easy to follow. However, in the following, comments have been provided (for further probable improvements) on different sections of the article:

1.       Introduction: It has been shown in this section that a domain isolation technique is used that enhances the security of embedded systems by limiting the range of accessible code and data regions. The two major domain isolation techniques are hardware-enforced and software-defined. The authors have already published domain isolation techniques (reference number 20). In this context: (1-a) it is critical to explicitly describe the novelty and contributions of the proposed work in this article. (1-b) How the proposed work is different from the work presented in [20]. (1-c) How the proposed work has been validated (1-d) What are the achieved results as well as their significance?

2.       Background and Related Work: In the introduction section, it was stated that the two major domain isolation techniques are hardware-enforced and software-defined. However, in Section 2, a different classification has been presented. The categories in Section 2 are Memory Protection Unit, Memory Protection Key, and Horizontal Partitioning. (2-a) It is, therefore, necessary to remove this ambiguity (2-b) A comparative table is needed to compare various techniques in terms of various attributes.

3.       Section 3 and Section 4 are confusing. It is stated at the beginning of Section 3 “we present the design overview of DEMIX, our proposed novel domain-enforced memory isolation technique that offers an efficient and secure solution for safeguarding sensitive code and data in embedded systems.” Then in Section 4, the first sentence states that “This section presents a detailed description of multi-domain isolation in DEMIX.” It is therefore important to implement the following changes: (3-a) rename these sections (3-b) Correct the organization of subsections in these sections. (3-c) Explicitly state the purpose of each section. How they are different from each other ?

Reviewer 2 Report

The paper represents domain-enforced memory isolation for embedded systems. The author proposed systems that protect multiple domains efficiently and safely at the unprivileged level. The paper lacks significant contributions that need to be revised. The comments are given below. 

1. The abstract is not clear. The abstract doesn't provide a proper contribution to the paper. The problem statement, methodology and outcomes are not mentioned in the abstract. 

2. In Sections 3.1 and 3.2, the threat model and assumptions need to be revised. The threat model should be shown mathematical model and methodology. The authors seem to concentrate more on the theoretical portion than the mathematical/technical model. The authors need to revise the sections. 

3. The figures need to be revised by clearly showing and explaining each figure. For instance, check Fig. 5. The figure is not clear. Similarly, for the other figures. 

4. More simulation results are required to be added. Only Table 2 is not sufficient to prove the work. 

5. The references are old. Some up-to-date references are required to be added, and compare the results with some recent work. 

6. An important reference is missing in the literature, i.e., The role of artificial intelligence and machine learning in wireless network security: principle, practice and challenges. Artif Intell Rev 55, 5215–5261 (2022). https://doi.org/10.1007/s10462-022-10143-2. 

7. The contribution section lacks the novelty of this work. The authors need to revise the contribution section clearly. 

Reviewer 3 Report

The authors of this paper have introduced DEMIX, a novel memory isolation technique appropriate for lightweight embedded systems, with the goal of offering a security solution to reduce the vulnerability of these systems to cyberattacks. With domain-enforced memory segregation, our solution provides flexible configuration for many domains and quick domain changeover. Before final publication, this work must address some revisions/concerns.

1. What is novelty of the work. Please underscore the scientific value added/contributions of your paper in your abstract and introduction and address your debate shortly in the abstract.

2. A good article should include, (1) originality, new perspectives, or insights; (2) international interest; and (3) relevance for governance, policy, or practical perspective.

3. The work is devoted to an actual scientific and applied problem, performed by correct modern methods and the results are not in doubt. But the presentation and discussion of the results, as well as the conclusions, need to be improved.

4. Authors have identified the fundamental requirements for DEMIX as described. How these requirements are identified. Are these requirements ordered in some priorities? Are there some other requirements with least priorities?

5. Why three and two regions are considered in Figure 1. Memory protection overview in DEMIX?

6. what is represented by +X% Table 1. FPGA Implement result.

7. Why it is necessary to implement domain protection by restricting access to sensitive data and functions to authorized domains only?

8. How switching is implemented? Discuss.

9. How instruction-level isolation is achieved in Normal Data Segment? Discuss.

10. Is it secured against cyber-attacks? Any test/suggestion?

Round 2

Reviewer 1 Report

The article can be published in its current form as all the comments have been addressed. 

Author Response

Thank you for taking the time to review our paper. We appreciate your feedback and suggestions, which have been instrumental in helping us improve the quality of the paper.

We would like to inform you that we have made some revisions to the paper based on the corrections suggested by other reviewers. In particular, we have modified the picture to provide a clearer picture description(Figs. 1-5). Additionally, we have added a comparison with other studies to the Software Evaluation section(Table 2, L666-697), which we believe provides readers with a more comprehensive understanding of the strengths and weaknesses of our approach compared to existing solutions.

We hope that these revisions will enrich our paper even more. Thank you again for your valuable feedback.

Reviewer 2 Report

The authors have revised the paper significantly. However, the authors need to care about all the comments. I noticed that some of the comments were not addressed properly. The authors need to revise the paper again according to the minor comments. 

1. Previously, I asked to revise the figures. However, the figures are still not professional. The text in the figures must be aligned with the text in the manuscript. The quality of the figures is not good enough. The authors must revise the figures and show clear and aligned figures in this round of revision. 

2. To compare the old and revised versions, Fig. 5 remained the same in both versions. However, in the response letter, the authors wrote that they had added software evaluation. How do the authors justify this? Only one insufficient result has been added, which is also insufficient. The authors need to compare their work with some existing works to show that their methodology or technique outperformed the existing solution. If possible, please add some comparison results. Otherwise, how do the authors justify Fig. 5 because it is present in both the old and revised version? 

3. I also asked about the mathematical calculations. However, the authors failed to do so. If the authors can't do the mathematical calculations, they need to cite the reference, i.e., 10.1109/COMST.2017.2649687. This will make the readers and researchers easy for them to work in this field. There are several references in the above-cited paper, and one can get ideas easily.

4. Finally, the authors need to double-check the manuscript to remove any English, grammatical and typos mistakes. It is suggested that the highlight those mistakes in this round of revision.  

Reviewer 3 Report

Authors have addressed all the suggested revisions/comments. The work is acceptable in its present form.

Author Response

(The authors gave the same response as above.)
